# Monitoring the after-effects of ischemic stroke through EEG microstates

**Fang Wang** [1]*, **Xue Yang**[1], **Xueying Zhang**[2], **Fengyun Hu**[3]*

1 West China Biomedical Big Data Center of West China Hospital, Sichuan University, Chengdu, China,
2 College of Information and Computer, Taiyuan University of Technology, Taiyuan, China, 3 Department of
Neurology, Shanxi Provincial People's Hospital Affiliated with Shanxi Medical University, Taiyuan, China

* wang_fang_ty@163.com (FW); fengyun71@163.com (FH)

## Abstract

### Background and purpose

Stroke may cause extensive after-effects such as motor function impairments and disorder of consciousness (DoC). Detecting these after-effects of stroke and monitoring their changes are challenging jobs currently undertaken via traditional clinical examinations. These behavioural examinations often take a great deal of manpower and time, thus consuming significant resources. Computer-aided examinations of the electroencephalogram (EEG) microstates derived from bedside EEG monitoring may provide an alternative way to assist medical practitioners in a quick assessment of the after-effects of stroke.

### Methods

In this study, we designed a framework to extract microstate maps and calculate their statistical parameters to input to classifiers to identify DoC in ischemic stroke patients automatically. As the dataset is imbalanced with the minority of patients being DoC, an ensemble of support vector machines (EOSVM) is designed to solve the problem that classifiers always tend to be the majority classes in the classification on an imbalanced dataset.

### Results

The experimental results show EOSVM get better performance (with accuracy and F1-Score both higher than 89%), improving sensitivity the most, from lower than 60% (SVM and AdaBoost) to higher than 80%. This highlighted the usefulness of the EOSVM-aided DoC detection based on microstates parameters.

### Conclusion

Therefore, the classifier EOSVM classification based on features of EEG microstates is helpful to medical practitioners in DoC detection with saved resources that would otherwise be consumed in traditional clinic checks.

Monitoring the after-effects of ischemic stroke
through EEG microstates. PLoS ONE 19(3):
e0300806. https://doi.org/10.1371/journal.
pone.0300806

Jiaotong-Liverpool University, CHINA

**Data Availability Statement:** All relevant data are
within the manuscript and its Supporting
information files.

**Funding:** There is no financial support in this work.

**Competing interests:** The authors have declared that no competing interests exist.

## Introduction

Stroke is one of the most prevalent neurological conditions worldwide and statistical data indicates that stroke ranks as the second leading cause of disability, constituting 11.4% of disabilities among affected individuals, following closely behind dementia in the elderly population [1]. It leads to several disabilities such as extensive motor function impairment, cognitive disorders, and disorder of consciousness (DoC) [2]. Detecting these after-effects of stroke and monitoring the changes in the clinical condition of patients are challenging jobs. They are currently undertaken via traditional behavioural and clinical examinations, which carry a high test-retest and inter-examiner variability [3]. For example, the National Institutes of Health Stroke Scale (NIHSS) score is mostly used to evaluate stroke-related neurologic impairment, the Glasgow Coma Scale (GCS), Coma Recovery Scale-Revised (CRS-R), et al, are usually used to assess the level of consciousness of patients. Any changes in the clinical state of a patient cannot always be identified promptly. Their detection largely relies on how long the interval is to the next clinical examination. Furthermore, significant manpower, time and other resources are consumed in these clinical examinations for both inpatients and outpatients. For stroke patients, pathological changes in certain areas of the brain can cause motor function impairment. Identification of motor disturbance in poststroke patients timely is necessary [4, 5].

Resting-state electroencephalography (EEG) monitoring provides an alternative way to potentially assist medical practitioners in a quick assessment of the after-effects of stroke. Although the spatial resolution of EEG is lower than images such as CT, PET and MRI, EEG is noninvasive, easy to monitor long-term and inexpensive, therefore, it is widely used in neurological examinations. Although the spatial resolution of EEG is not as good as images such as CT, PET and MRI, EEG recordings always have higher time domain resolution [6]. Quantitative EEG indices of sub-acute ischaemic stroke were correlated with NIHSS scores and may inform future management of stroke patients [7]. The investigation of the correlation between early EEG biomarkers and functional and morphological outcomes in thrombolysis-treated strokes helps better establish the treatment strategies [8]. A recent EEG fractal analysis study shows that the stroke throughout the acute and early subacute stages show significantly less complex brain activity compared to healthy subjects [9].

Several previous studies demonstrated that there are brief periods in global electrical brain activities on the scalp that remain semi-stable [10], namely microstates. Microstates analysis segments EEG data into a limited number of clusters with a duration varying between 40–150ms based on global points [11]. These transient periods of stability have different topographical representations, namely microstates. EEG Microstate analysis has been increasingly investigated for the spatial and temporal properties of whole-brain neuronal networks [11].

As a tool for the study of brain activities, EEG Microstates have been applied in the investigation of neuropsychiatric diseases. A study in NeuroImage examined the degree to which spatial and temporal properties of microstates might be altered by manipulating cognitive tasks (a serial subtraction task versus wakeful rest). It provided visual information (eyes open versus eyes closed conditions) to medical practitioners [10]. Another study demonstrated that altered states of consciousness, e.g., sleep, hypnosis, and meditation, were correlated with changes in microstate properties [12]. Other studies explored EEG microstate changes in neurological diseases, such as schizophrenia [13–15], head injury [16], dementia [14], and narcolepsy [17].

Many investigations have been undertaken on how the properties of EEG microstates vary across different cognitive tasks, genders, medications, and diseases. Despite these efforts, the challenge of microstate analysis is to design experiments that are capable of establishing direct causal relations between the EEG microstates and certain hypotheses [11]. Particularly, only one report has been found so far on the examination of EEG microstates in stroke patients

[18]. It compared the parameters of EEG microstates derived from 47 stroke patients and 20 healthy controls in three groups (the left hemisphere lesion stroke group, the right hemisphere lesion stroke group, and the healthy controls group). These previous studies have shown the potential of EEG microstates to extract characteristics from patients with stroke.

Our work in this paper aims to investigate the correlation between EEG microstates with the clinical states of stroke patients, such as the level of consciousness, the existence of motor disturbance, and the side of the motor disturbance. In addition, we employed two widely accepted classifiers (a single SVM classifier and an AdaBoost classifier) as base models to build the ensemble of classifiers to classify the stroke patients with DoC and those without DoC.

Our work in this paper makes two main contributions:

1. The first contribution is our establishment of correlations between EEG microstates and the clinical states of stroke patients through experimental studies of 152 patients. We designed an experimental procedure to extract microstate maps from a single dataset aggregated from multiple EEG datasets of all patients. Then, we investigated the correlations between EEG microstates with the level of DOC (awake, somnolence, stupor, light coma, middle coma, and deep coma) through the Spearman correlation coefficient. Moreover, a one-way ANOVA analysis was carried out to investigate the differences of EEG microstate parameters between different motor disturbance groups.

2. The second contribution is our design of a classifier for the detection of the DoC of stroke patients. We employed an ensemble of support vector machines (EOSVM) as the framework of the classifier. EEG microstates together with the statistical microstate parameters were input to the classifier. We compared the performance of the EOSVM under different numbers of microstate maps and different settings of the EOSVM majority voting.

The rest of this paper is organized as follows: Materials and Methods Section presents materials, procedures of EEG microstate analysis, and the framework of our EOSVM classifier. Our main results are summarized in the Result Section on correlation analysis between microstates and level of DoC in stroke patients, and comparing classification results from EOSVM with Support Vector Machine (SVM) and Adaptive Boosting (AdaBoost). Further discussions on the experimental results are given in the Further Discussions Section. Finally, the Conclusion Section concludes the paper.

## Materials and methods

### Participants

There were 152 stroke patients (mean age = 64.76 years, standard deviation (SD) = 15.63 years) in this study. These subjects were patients admitted to the neurology department at Shanxi Provincial People's Hospital after acute stroke from 2017 to 2018. Table 1 summarized the demographics and clinical characteristics of the participants. Among the 152 stroke patients, the Inclusion criteria were as follows: (1) The patients were diagnosed with ischemic stroke, (2) EEG data were recorded and available for analysis, and (3) the corresponding assessment of consciousness by medical practitioners was recorded. Exclusion criteria were: (1) patients younger than 18 years old, and (2) pregnant patients. All data were collected as part of a prospective observational cohort study approved by the local institutional review board of the Shanxi Provincial People's Hospital and informed consent was obtained from all subjects or their legal guardian. All methods in this study were carried out in accordance with relevant guidelines and regulations.

The state of consciousness of each patient was determined by using a hierarchical battery of observation assessments. The assessment method of consciousness has been summarized in

**Table 1. Demographics and clinical characteristics of 152 patients.**

| State of Consciousness | Number of Patients | Female Patients | Age (Mean, SD) |
|---|---|---|---|
| Awake | 95 | 24 | 63.05, 15.31 |
| Somnolence | 29 | 9 | 69.17, 18.17 |
| Stupor | 14 | 5 | 65.29, 11.22 |
| Light coma | 7 | 2 | 70.71, 16.64 |
| Middle coma | 6 | 2 | 60.33, 13.94 |
| Deep coma | 1 | 0 | 76.00, 0.00 |
| OVERALL | 152 | 42 | 64.76, 15.63 |

our previous study [19]. For all patients, neurological examinations with the assessments of consciousness were performed before EEG signals were recorded. All clinical assessments were performed by medical practitioners who were blinded to the EEG measures of the patients.

## EEG data acquisition and pre-processing

EEG signals were recorded through a bedside digital video EEG monitoring system (Solar 2000 N, Solar Electronic Technologies Co., Ltd, Beijing, China) at a sampling rate of 100 Hz. The specific electrodes sites were positioned at FP1, FP2, C3, C4, O1, O2, T7, T8, A1, and A2 according to the international 10–20 system. EEG data were recorded continuously for at least 2 hours and impedance was established below 10kΩ for all electrode sites. The maximum allowed interval between the end of the clinical examination and the start of the EEG recording time was 30 minutes in this study.

Our EEG pre-processing was carried out offline in MATLAB (Mathworks, Natick, MA) with the EEGLAB toolbox (version 14.1.1b). The continuous EEG data were first re-referenced to an average reference, before which the bad channel was rejected. Then, high-pass filtering (0.5 Hz) and low-pass filtering (40 Hz) were successively applied to the EEG signals by using a basic finite impulse response (FIR) filter. For the resulting EEG signals, we detected artefacts based on eye movements, muscle activity, and amplitude threshold violations (150 $\mu$V). The trials with any of these artefacts were removed [20, 21].

## Microstate analysis

Our microstate analysis followed the procedure in Fig 1. Briefly speaking, the clean and processed EEG data recorded in the first five minutes were included in the microstate analysis for each subject. The EEG datasets from all 152 stroke subjects were aggregated into one dataset. This dataset was then used to derive microstate prototypes. After that, these microstate prototypes were back-fit to EEG data from each subject. After the EEG microstates were defined for each of the patients, statistical temporal parameters were calculated from the derived datasets. The microstates in this study were extracted through a plugin for EEGLAB, Microstate EEGLAB toolbox [22].

**Microstate segmentation.** Here, we explain how to derive microstate prototypes from the aggregated dataset. Firstly, the global field power (GFP), which is the spatial standard deviation of EEG signals across all channels, was calculated. Then, a clustering method was used to group the GFP sequences into a small set of classes based on topographic similarity. We used a modified k-means method in this study. Each of the resulting clusters could describe a topographical prototype, namely a microstate prototype.

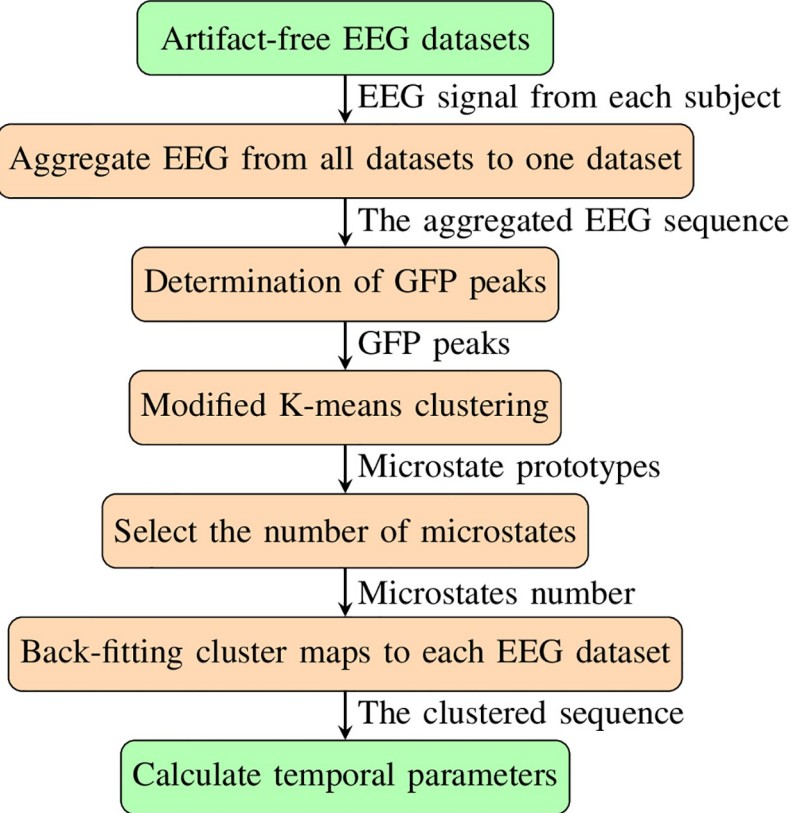

**Fig 1. The procedure of the microstate analysis in this study.**

The Modified k-means method adds several features to clustering [22]. Here the modified k-means models introduce two differences compared to conventional k-means. The first is that the topographical maps of prototypical microstates with opposite polarity are assigned to the same cluster. The second difference is that modified k-means models the activations of the microstates i.e. models the strength of the microstates for each time point [22].

**Microstate parameters.** Microstate parameters, duration, occurrence, coverage, and variance were calculated in the following for the sorted microstates [10, 23]:

1. Duration: the average duration that a given microstate remains stable.

2. Occurrence: the number of times a microstate occurred during one second.

3. Coverage: the total percentage of time covered by a given microstate.

4. Variance: global explained variance (GEV) is a measure of how similar each EEG sample is to the microstate prototype that the sample has been assigned to [22]. The higher the GEV is, the higher the similarity is.

## Classification framework

In general, datasets from a hospital, such as EEG signals, are imbalanced. For EEG signals from stroke patients, the datasets consist of much more wakeful samples than DoC ones. A common problem in training a classifier from imbalanced datasets is that the trained classifier

is more likely to predict a sample as the majority class. This will lead to critical false negatives when DoC is considered positive.

To address this problem, we designed a classifier of an ensemble of support vector machines (EOSVM) formed from multiple SVM classifiers, as shown in Fig 2. In the training phase, the overall datasets for training are aggregated and then split into $N$ subsets. These $N$ subsets are fed into $N$ SVMs, respectively. In the test phase, all SVMs take the same input for prediction. Each SVM makes its own decision, which classifies the corresponding sample into wakefulness or a DoC state. After that, multiple predictions from the $N$ SVMs are fused through a voting rule to make the final prediction. In this study, we use a simple majority voting rule.

The number of SVMs in the EOSVM classifier should be adjusted according to the distribution of a dataset. Normally, the more heavily imbalanced the dataset is, the more SVMs should be used in the EOSVM classifier. However, the largest number of SVMs in EOSVM should be capped by the number of combinations of $r$ objects chosen from $n$ objects as follows:

$$C(n, r) = \frac{n!}{r!(n-r)!} \tag{1}$$

where $r$ represents the number of the minority classes in the training dataset, and $n$ represents the number of the majority classes. In this study, the minority classes are the DoC classes, while the majority classes are the awakeness classes.

The classification of the EEG signals is performed in three steps: 1) data aggregation and splitting, 2) classifier training (in the training phase) and prediction, and 3) voting for final prediction. These steps are already shown in Fig 2. They are described below in more detail.

**Data aggregation and splitting.** We randomly split the entire dataset into a training set (80% of the data) and a test set (20% of the data). In order to train the $N$ SVMs in the EOSVM classifier, we further build $N$ balanced training subsets from the training set. For each subset, we first include all the DoC samples from the training set since they only account for a small

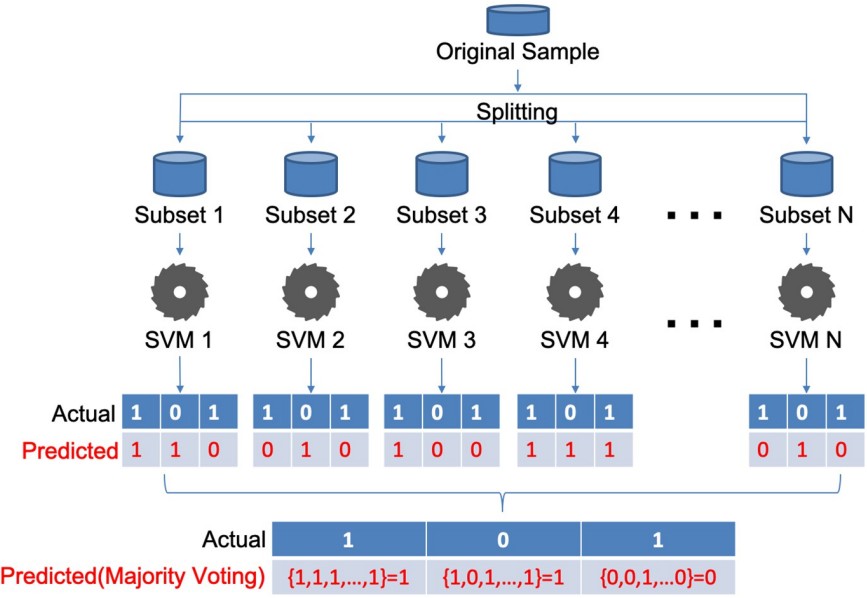

**Fig 2. The framework of our EOSVM classifier.** In the training phase, each of the SVMs in the EOSVM takes a different subset of data as its input as shown in the upper part of the figure. In the test phase, all SVMs take the same input for prediction, for which an example of 3 samples is shown in the lower part of the figure.

proportion of the training set, and then randomly select the same number of wakefulness samples from the training set to add to the training subset. Each training subset and the test set make up a subset as an input to an SVM.

**Classifier training and prediction.** To train each of the $N$ SVM classifiers, we employ the Gaussian kernel function. In the training, each SVM automatically tunes the capacity of the classification function by maximizing the margin between training examples and class boundary, optionally after removing some atypical or meaningless examples from the training data. The hyper-parameters are obtained by maximizing the marginal.

For prediction, the $N$ SVMs make their respective decisions to diagnose in or out of either wakefulness or DoC of the patients.

**Voting for final prediction.** The $N$ predictions from the $N$ SVMs may be the same or different. They are fused through a voting rule. In standard majority voting, the two classes with the most votes from the $N$ SVMs is determined as the final classification result. However, if the votes for class 1 are only more than the votes for class 2 a little, e.g., 51 versus 49, we only have low confidence for the classification result of the voting result. If the votes for one class are much higher than the other, e.g., 88 versus 11, the final classification is more clinically reliable. Therefore, only exploring the voting threshold of 50% is not enough. We tested over 50% majority voting up to 100% majority voting in the experiments. From our test experiments, some insights were developed on the number of EEG microstates for the best prediction performance, and the threshold of the majority voting for the best prediction results.

## Evaluation of classification performance

To evaluate the performance of classification and prediction, we use two groups of metrics. The first group is introduced from machine learning and classification perspectives, while the second group is more on medical assessment.

The first group of metrics includes Accuracy, F1-Score, Sensitivity, and Specificity, which are common metrics for classification assessment in machine learning. We consider subjects with DoC as positive cases in this study.

The physical meanings of the four metrics (Accuracy, F1-Score, Sensitivity, and Specificity) are described below. The performance of accuracy characterizes the percentage of true classifications of overall classified subjects. Sensitivity refers to the ability to identify true positives among all subjects who have been predicted as positive, i.e., the patients who are diagnosed with DoC do have DoC. Specificity quantifies the ability to detect true negatives among all subjects who have been predicted as negative, i.e., the patients diagnosed with no DoC do have no DoC. It helps rule out those patients without DoC. Moreover, F1-Score is the harmonic mean of precision and sensitivity.

While the above four metrics in the first group are useful in evaluating the overall performance of the EOSVM performance, clinical assessments of patients require more certain results. For example, given a classification accuracy of 90%, what can we say if a patient is predicted by the classifier to have DoC? Likely, the patient has the DoC. However this is not enough in a medical assessment. We need to know if the patient really has DoC or not.

Therefore, three additional metrics are designed to characterize the capability of the EOSVM classifier to predict assessment results more relevant to clinic diagnosis. They form the second group of our metrics. They are specified in this study as True Prediction, False Prediction, and Not Sure (for those cases the EOSVM classifier is unable to give a certain prediction useful for clinical examinations).

Before calculating True Prediction, False Prediction, and Not Sure, we need to define Prediction Rate as a metric to evaluate the percentage of subjects with a prediction among all

subjects. It is formulated as:

$$PredictionRate = \frac{TP + FP + TN + FN}{N_{awa} + N_{DoC}} \times 100\% \qquad (2)$$

where $N_{DoC}$ and $N_{awa}$ are the total number of positive subjects (DoC) and negative subjects (no DoC), respectively.

Then the second group of metrics are mathematically described as:

$$TruePrediction = Accuracy \times PredictionRate \qquad (3)$$

$$NotSure = (1 - PredictionRate) \times 100\% \qquad (4)$$

$$FalsePrediction = (1 - TruePrediction - NotSure) \times 100\% \qquad (5)$$

It is worth mentioning that the sum of TP, TN, FP, and FN is not equal to $N_{awa} + N_{DoC}$. This is because some subjects may have no final predictions for some settings of the majority voting in EOSVM.

## Statistical analysis

Firstly, Spearman correlations between the level of consciousness and microstate parameters were employed to explore the relationship between consciousness and microstates. Secondly, an independent T-test was used to determine significant differences between different motor disturbance groups for investigating the relationships between motor impairment and micro-states. Finally, we analyzed the statistics of the classification results.

## Results of statistical analysis

Here, we analyzed more than four primary microstates prototypes coming from the clustering. For the number of clusters of microstates, previous experimental and clinical studies focused on four primary cluster maps, which were labelled as classes A, B, C, and D [10, 12, 17, 24–26]. Ideally, the best number of clusters should be estimated for each dataset individually using robust optimization criteria, rather than a fixed value [11, 22]. Considering the extracted microstates in our dataset, we investigated various scenarios with a varying number of micro-state maps. For six microstates, we labelled them as A, B, C, D, E and F as shown in Fig 3. These six microstates were also explored by previous studies [11, 27]. Additionally, we also explored the scenarios when EEG data were clustered to 2, 4, 6, 8, 10, and 12 microstate maps, as shown in Fig 4.

## Microstates and the level of consciousness

To explore whether the microstate parameters (duration, occurrence and coverage) can be used as features to classify the different levels of consciousness, we analyzed the correlation between microstates parameters with the level of consciousness. In clinic examinations, there are six levels of consciousness (L6), i.e., wakefulness, somnolence, stupor, light coma, middle coma, and deep coma. They are numbered from 1 to 6, respectively. The bigger the numerical value of the level is, the worse the state of consciousness is. Spearman correlation coefficients are calculated between microstate parameters and the level of consciousness. The results are shown in Fig 5 and the Table 2.

The first columns of both correlation matrices in Fig 5 reveal the relationships between 18 EEG features and the level of consciousness. The 18 EEG features are the three microstate

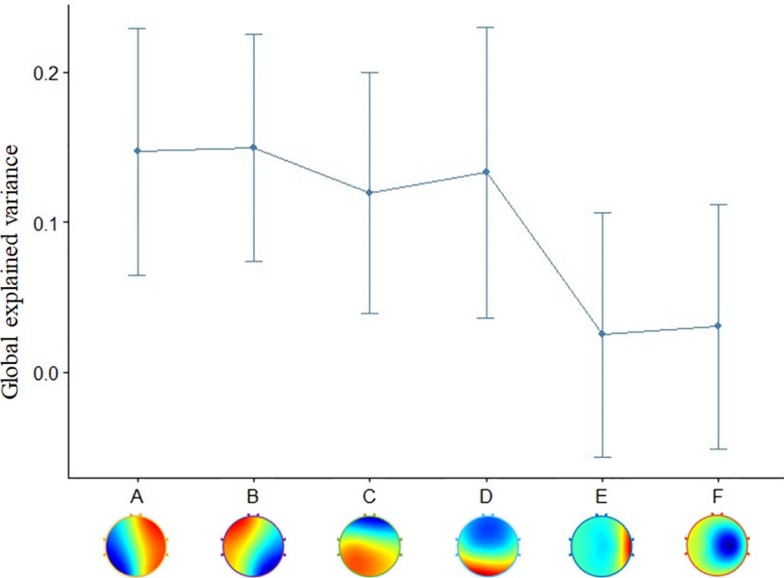

**Fig 3. The figure shows the six microstate prototypes of A, B, C, D, E and F at the bottom of the figure.** In the top part of the figure, the line shows the mean and standard deviation of Global explained variance (GEV) for the six microstates (A, B, C, D, E and F). GEV is a measure of how similar each EEG sample is to the microstate prototype it has been assigned to [22]. In the correlation analysis and T-test in this study, we focused on these 6 cluster maps as in some previous studies.

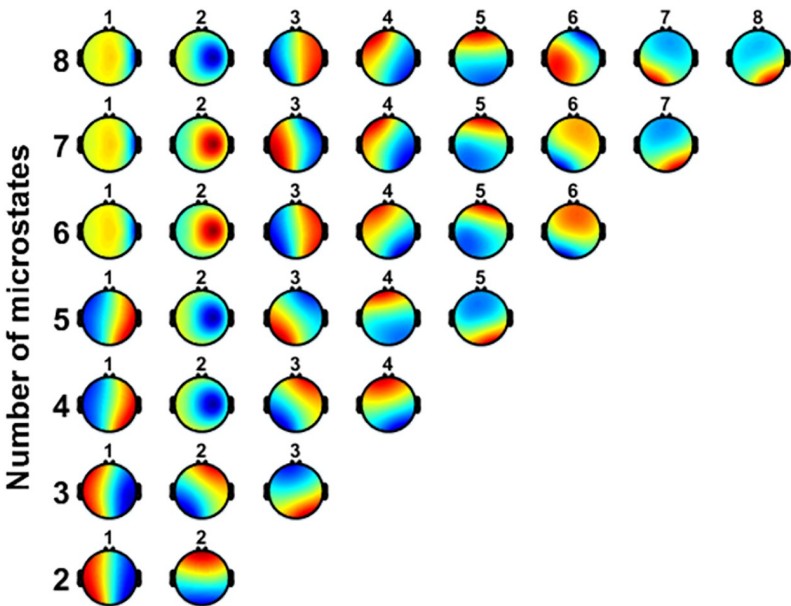

**Fig 4. Microstate prototypes of spatial clustering analysis using a modified K-means clustering method.** The clustering analysis of the maps was carried out at the GFP peaks of the EEG dataset aggregated from all EEG files of 152 subjects. The graph shows the cluster maps when EEG data were clustered to $k$ microstates for $k = 2, 3, \cdots, 8$.

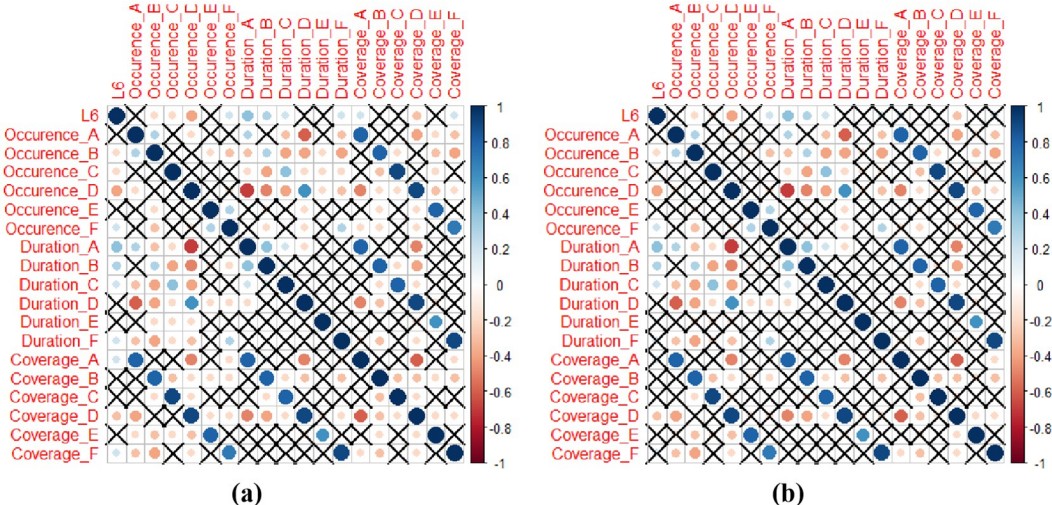

**Fig 5. Correlation matrices visualized with coloured significance levels of correlations between EEG microstates and the level of consciousness in stroke subjects.** The colour 'blue' represents positive correlations and the color 'red' refers to negative correlations. In subfigure (a), correlations with $p$-value <0.05 are considered to be significant and the insignificant ones are marked with '×'. Subfigure (b) are correlations with $p$-value <0.01.

parameters, duration, occurrence and coverage from the six different microstates A, B, C, D, E and F. Let us focus on the four microstates A, B, C, and D first, which have been previously examined in the literature. It is seen from Fig 5 that the 'occurrences' of microstate B ($P < 0.01$), C ($P < 0.05$), and D ($P < 0.01$) were significantly correlated with the level of consciousness in stroke patients. For parameter 'duration', microstates A, B, and C had significant correlations with the level of consciousness with a positive trend ($P < 0.01$). The 'coverage' of microstates A ($P < 0.05$) and D ($P < 0.01$) were also significantly correlated with the level of consciousness in stroke subjects (with positive and negative correlations, respectively).

If we focused on microstates E and F, it was observed from Fig 5 that microstate E did not show significant correlations with the level of consciousness in all the three statistical parameters 'occurrence', 'duration' and 'coverage'. However, microstate F was significantly correlated with the level of consciousness with a positive trend in all three statistical parameters.

**Table 2. Pearson correlation tabular results.**

| Pearson r | r | R squared | P (two-tailed) | P value summary | Significant? (alpha = 0.05) |
|---|---|---|---|---|---|
| L6 vs. Occurence_B | -0.2463 | 0.06068 | 0.0022 | ** | Yes |
| L6 vs. Occurence_C | -0.1792 | 0.03213 | 0.0271 | * | Yes |
| L6 vs. Occurence_D | -0.3808 | 0.145 | <0.0001 | **** | Yes |
| L6 vs. Occurence_F | 0.2198 | 0.0483 | 0.0065 | ** | Yes |
| L6 vs. Duration_A | 0.4472 | 0.2 | <0.0001 | **** | Yes |
| L6 vs. Duration_B | 0.2969 | 0.08817 | 0.0002 | *** | Yes |
| L6 vs. Duration_C | 0.2227 | 0.04958 | 0.0058 | ** | Yes |
| L6 vs. Duration_F | 0.1682 | 0.02828 | 0.0384 | * | Yes |
| L6 vs. Coverage_A | 0.189 | 0.03573 | 0.0197 | * | Yes |
| L6 vs. Coverage_D | -0.2874 | 0.08259 | 0.0003 | *** | Yes |
| L6 vs. Coverage_F | 0.1996 | 0.03985 | 0.0137 | * | Yes |

Table 2 presents the results of Pearson correlation analysis corresponding to Fig 5, outlining the correlation coefficients among 12 significant microstate parameters out of the total 18 features investigated. The table illustrates the Pearson correlation coefficient (r) values alongside their squared correlation (r squared), two-tailed p-values, and the summary of significance levels (using alpha = 0.05). Each correlation entry signifies the relationship strength between microstate parameters and the level of consciousness. Significant correlations, denoted by asterisks (*), ***, or ****, indicate the respective p-values being below the threshold of 0.05, highlighting statistically significant associations between these microstate parameters.

## Microstates and motor impairment

We investigated whether there were any statistically significant differences in microstate parameters among four groups through the one-way analysis of variance (ANOVA). The four groups are the left motor disturbance (41 subjects), the right motor disturbance (45 subjects), the bilateral motor disturbance (6 subjects), and the no motor disturbance (57 subjects). Among the total 152 subjects in this study, 3 subjects have no information about motor impairment. Thus, 149 subjects are included in the motor impairment analysis. Table 3 revealed the mean and standard deviation (SD) of the microstates parameters. It also showed the results of the one-way ANOVA and the following post hoc test determining which of the four groups differ from each other.

For the microstate parameter occurrence rate of microstate A, the result of one-way ANOVA found that there was a statistically significant difference between groups, $F(3, 145) = 4.411$, $p = 0.005 < 0.05$. An LSD post hoc test revealed that the occurrence of microstate A was

**Table 3. Microstate characteristics, between-group ANOVA (df = 3), and multiple comparisons (LSD post hoc test).** Significant differences ($p < 0.05$) are displayed in bold type.

| Microstate | Descriptives (Mean, SD) | | | | ANOVA | | Post-hoc comparisons -p value | | | |
|---|---|---|---|---|---|---|---|---|---|---|
| | No(57) | Left(41) | Right(45) | Bilateral(6) | F | p | L vs R | N vs L | N vs R | N vs B |
| Occurrence | | | | | | | | | | |
| A | 4.59, 0.90 | 4.37, 1.22 | 5.10, 0.82 | 4.91, 0.93 | 4.411 | **0.005** | **0.001** | 0.272 | **0.010** | 0.444 |
| B | 4.76, 0.96 | 4.56, 1.04 | 4.97, 0.78 | 4.88, 0.66 | 1.457 | 0.229 | **0.040** | 0.273 | 0.268 | 0.776 |
| C | 3.78, 1.05 | 3.65, 0.84 | 3.90, 0.90 | 4.04, 0.41 | 0.626 | 0.599 | 0.230 | 0.492 | 0.550 | 0.532 |
| D | 4.08, 1.24 | 4.59, 1.42 | 4.33, 1.19 | 4.59, 1.55 | 1.352 | 0.260 | 0.351 | 0.055 | 0.330 | 0.361 |
| E | 1.61, 0.85 | 1.69, 0.81 | 1.64, 0.84 | 1.33, 0.57 | 0.361 | 0.781 | 0.771 | 0.618 | 0.844 | 0.432 |
| F | 2.14, 0.96 | 1.93, 0.83 | 1.96, 0.74 | 1.70, 0.28 | 0.896 | 0.445 | 0.869 | 0.226 | 0.287 | 0.227 |
| Duration | | | | | | | | | | |
| A | 49.64, 9.54 | 49.17, 12.67 | 49.40, 7.90 | 50.51, 11.67 | 0.039 | 0.990 | 0.915 | 0.819 | 0.905 | 0.843 |
| B | 50.94, 9.95 | 47.73, 7.71 | 47.64, 5.96 | 46.96, 3.91 | 1.948 | 0.124 | 0.958 | 0.056 | **0.044** | 0.256 |
| C | 47.53, 10.96 | 44.98, 7.48 | 43.00, 7.22 | 44.69, 7.16 | 2.215 | 0.089 | 0.305 | 0.164 | **0.012** | 0.459 |
| D | 46.88, 9.53 | 51.71, 10.76 | 45.41, 9.17 | 48.21, 13.69 | 3.141 | **0.027** | **0.004** | **0.019** | 0.460 | 0.757 |
| E | 35.24, 8.37 | 35.15, 6.80 | 39.26, 44.25 | 31.16, 4.81 | 0.352 | 0.788 | 0.451 | 0.986 | 0.425 | 0.706 |
| F | 36.89, 8.14 | 39.54, 27.16 | 33.20, 4.03 | 32.29, 1.83 | 1.395 | 0.247 | 0.057 | 0.398 | 0.229 | 0.485 |
| Coverage | | | | | | | | | | |
| A | 0.23, 0.07 | 0.22, 0.09 | 0.26, 0.07 | 0.25, 0.10 | 1.830 | 0.144 | **0.034** | 0.574 | 0.085 | 0.443 |
| B | 0.24, 0.08 | 0.22, 0.07 | 0.24, 0.06 | 0.23, 0.04 | 1.007 | 0.392 | 0.230 | 0.092 | 0.663 | 0.619 |
| C | 0.18, 0.07 | 0.17, 0.05 | 0.17, 0.06 | 0.18, 0.04 | 0.730 | 0.536 | 0.714 | 0.169 | 0.309 | 0.921 |
| D | 0.20, 0.08 | 0.25, 0.12 | 0.20, 0.10 | 0.24, 0.15 | 2.377 | 0.072 | **0.049** | **0.013** | 0.668 | 0.353 |
| E | 0.06, 0.05 | 0.06, 0.04 | 0.06, 0.05 | 0.04, 0.03 | 0.279 | 0.841 | 0.983 | 0.899 | 0.915 | 0.400 |
| F | 0.08, 0.06 | 0.08, 0.09 | 0.07, 0.03 | 0.06, 0.01 | 0.937 | 0.425 | 0.264 | 0.906 | 0.183 | 0.289 |

statistically significantly lower in the left motor disturbance group (4.37 ± 1.22, *p* = 0.001) and no motor disturbance group (4.59 ± 0.90, *p* = 0.010) compared to the right motor disturbance group (5.10 ± 0.82) which are shown in Table 3.

For the microstate parameter duration of microstate D, the result of one-way ANOVA found that there was a statistically significant difference between groups, F(3, 145) = 3.141, *p* = 0.027 < 0.05. An LSD post hoc test revealed that the duration of microstate D was statistically significantly higher in the right motor disturbance group (45.41 ± 9.17, *p* = 0.004) and no motor disturbance group (46.88 ± 9.53, *p* = 0.019) compared to the left motor disturbance group (51.71 ± 10.76) as shown in Table 3.

## Results of classification

In this study, we considered classifying a subject into one of two classes, the awake and somnolent subjects (124) as class 1, and subjects with stupor and coma (28) as class 2. We also classified the awake (95) and 57) DoC classes, however, the classification performance was not good (the highest accuracy was below 80%). Therefore, the following results shew the classification of DoC patients (consisting of stupor, light coma, middle coma, and deep coma) and awake controls (consisting of wakefulness and somnolence).

In our classification, the inputs to each classifier included four microstate parameters 'duration', 'occurrence', 'coverage' and 'GFP' as classification features from each microstate. The state of consciousness (DoC or wakefulness) was also inputted as a label to the classifier.

In the following, we will show the classification results from six well-developed classifiers as benchmark classifiers. The six existing classifiers are Support Vector Machine (SVM), K-Neighbors (KNC), Decision Tree (DTC), Random Forest (RF), Ada Boost (ABC) and Bagging (BC). Then, we will present the classification results from our ensemble classifier EOSVM.

### Classification results from existing classifiers

**Classification results from SVM.** First of all, we use a single SVM as a benchmark to classify the positive cases (DoC subjects) and negative cases (no DoC subjects) on three datasets: the original dataset, random over-sampling dataset and random under-sampling dataset. To achieve a stable result, it is executed 30 times for features from a different number of microstates (2, 4, 6, 8, 10, 12).

Fig 6 shows the accuracy, sensitivity, specificity and F1-score from the classifier SVM. The results of the original dataset are depicted in blue box plots. It is seen from these box plots that the overall specificity for the different number of microstate features is around 90%, which is reasonably good. However, the results of sensitivity and F1-Score for the different number of microstate features are under 50% and 60%, respectively, which are poor for a clinically confident diagnosis decision. Thus, the classification over the original samples with heavy imbalance is not acceptable from the benchmark SVM classifier.

One may think that the poor performance in sensitivity and F1-score results from data imbalance. We remove data imbalance in the original training set through random under-sampling and random oversampling, respectively. Random under-sampling reduces the number of majority-class subjects to match the minority-class count. Random over-sampling increases the number of minority-class subjects to match the majority-class count. Then, we evaluate the classification performance of the SVM trained with the under-sampled and over-sampled datasets, respectively.

The classification results of 30 runs for features from the different numbers of microstates are shown in the green and orange box plots of Fig 6. The orange box plots are derived from

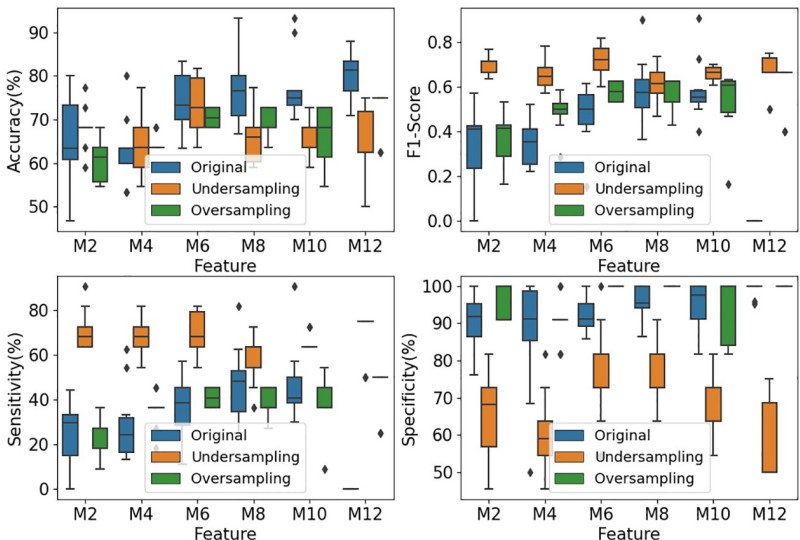

**Fig 6. Classification results from SVM.** *M+ number* on the horizontal axis refers to how many microstates maps the features are derived from. For example, M2 refers to the features derived from 2 microstate maps.

under-sampling, while the green box plots are obtained from over-sampling. It is seen from these green and orange box plots that

- Random under-sampling does improve the performance of F1-score and sensitivity; and

- Random over-sampling improves the specificity performance.

Although the performance of sensitivity and F1-Score from under-sampled datasets has improved significantly, the accuracy from random under-sampling is statistically poor, mostly below 70%. Therefore, the SVM classifier does not give satisfactory classification results over the balanced data constructed from the original imbalanced data through either random under-sampling or random over-sampling. Further improvement is required, and will be achieved through our EOSVM.

**Classification results from AdaBoost.** As AdaBoost is one of the most efficient machine learning methods in recent years [28, 29], we employ classifier AdaBoost as another benchmark to classify the positive cases (DoC subjects) and negative cases (no DoC subjects) in our experiments. Similarly, the classifier is executed 30 times to get a stable result and the three datasets (the original dataset, random over-sampling dataset and random under-sampling dataset) are investigated.

Fig 7 shows the accuracy, sensitivity, specificity and F1-score from the classifier AdaBoost. The results of the original dataset are depicted in the blue box plots. From these box plots, high specificity, low sensitivity, and low F1-Score are observed, which are similar to those in the benchmark classifier SVM in Fig 6. Even worse, the sensitivity from most microstate features tends to zero in some runs, implying that the benchmark classifier AdaBoost classifies all samples to the majority class, i.e., negative cases.

Also, the random under- and random over-sampled datasets are respectively classified by using AdaBoost. The classification results of 30 runs are shown in the green (random over-sampling) and orange (random under-sampling) box plots of Fig 6. It is seen from these green and orange box plots that

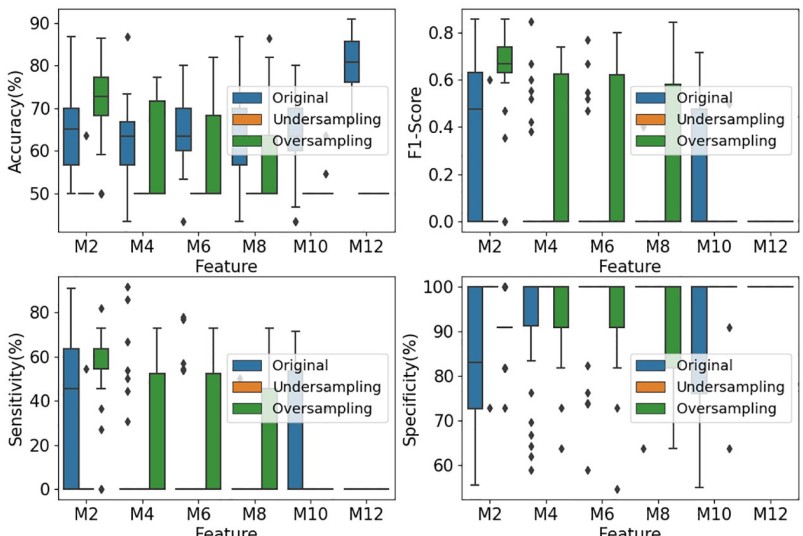

**Fig 7. Classification results from AdaBoost.** *M+ number* on the horizontal axis refers to how many microstates maps the features are derived from. For example, M4 refers to the features derived from 4 microstate maps.

- Random under-sampled datasets lead to the predictions of all the cases to be in the majority class (i.e., negative cases; and

- Random over-sampling improves the performance of the F1-score and sensitivity.

However, the improved sensitivity performance is still statistically poor, mostly below 60%. Also, the accuracy is sacrificed when AdaBoost is trained with either random under-sampling or random over-sampling. From these results, it is seen that the classifier AdaBoost predicts samples to be in the majority class more seriously than the classifier SVM does. Therefore, AdaBoost does not perform better on our imbalanced dataset than the classifier SVM does.

**Classification results from other existing classifiers.** To further explore the performance of existing classifiers on our dataset, we used the other four classifiers: DTC, KNC, BC, and RF, to classify subjects with DoC and without DoC.

To ensure stable classification operations, we used the stratified 5-fold cross-validation method to train the existing classifiers. This method, widely used in classification, ensures that the proportions between classes are the same in each fold as they are in the whole dataset.

The 5-fold cross-validation classification results from the existing classifiers are shown in Fig 8. From the figure, it can be seen that the mean accuracy and F1-Score from the six classifiers are all below 80%. Although the mean specificity is good, almost 90%, the means of sensitivity from all six classifiers are below 65%. Sensitivity representing how many subjects with DoC are predicted correctly is the most critical evaluation metric in this study.

## Classification results from EOSVM

For the classification from EOSVM, we present the results from EOSVM consisting of 100 SVMs because the performance under this setting is better than that under other settings of 10, 20, 50, and 150 SVMs in our experiments. When EOSVM is trained, each SVM in the EOSVM gives a prediction. Altogether, 100 predictions are obtained from the 100 SVMs of EOSVM. These 100 predictions may be the same or different. Thus, they are fused to give the final

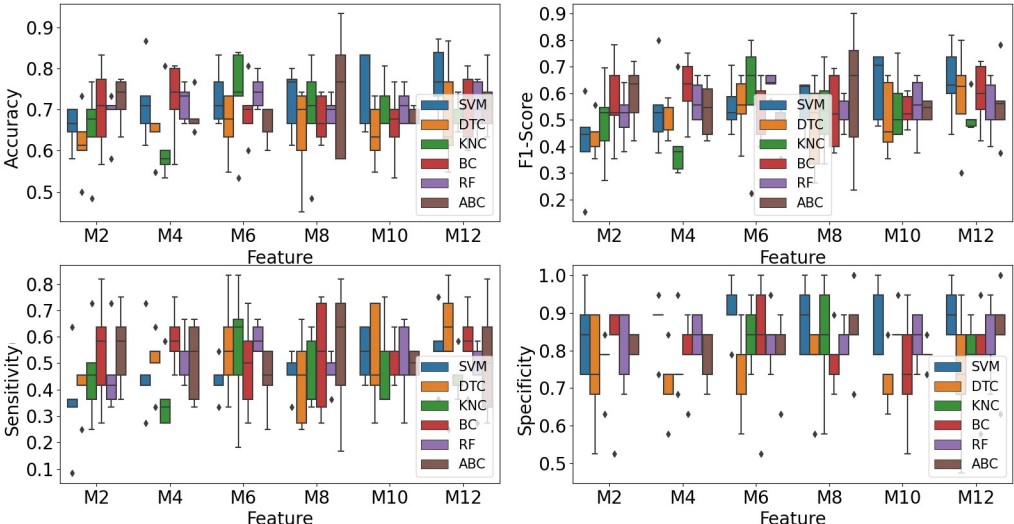

**Fig 8. Classification results from 6 classifiers.** *M + number* on the horizontal axis refers to how many microstate maps the features are derived from. For example, M4 refers to the features derived from 4 microstate maps.

prediction. In this study, the fusion is carried out through simple majority voting. Different settings of the majority voting are tested ranging from over 50% up to 100% majority voting.

**EOSVM classification result based on the first group of evaluation metrics.** The impact of the number of microstate maps on the performance of the EOSVM classification was investigated. The results were used to develop a guideline for the selection of subject features to classify subjects to DoC or wakefulness.

The EOSVM was trained with different numbers of microstate features. In this study, the settings of 2, 4, 6, 8, 10 and 12 microstate maps were considered for EOSVM training. For each of these settings, the EOSVM classifier was trained 30 times. The classification results from the trained EOSVM are depicted in Fig 9. The blue, orange, and green box plots refer to the classification results of EOSVM from 50% majority voting, 90% majority voting, and 100% majority voting, respectively.

It is observed from the green box plots in Fig 9 that with the requirement of a consensus from all SVMs (i.e., 100% majority voting), the classification accuracy in the scenario of 6, 8 and 10 microstates all give about 89% accuracy of classification and prediction. Meanwhile, the F1-Score from 6 microstate maps and 10 microstate maps are around 89%. These performance metrics have been improved significantly in comparison with the results from a single SVM classifier (Fig 6) and AdaBoost classifier (Fig 7).

However, what is the percentage of the patients that can be diagnosed in or out of DoC with an accuracy of about 89%? This is another issue that can be captured by our second group of three metrics defined in Eqs (3)–(5). This will be investigated later in the Materials and Methods Section.

The orange box plots in Fig 9 illustrate the classification results when a 90% majority voting rule is employed, which means that at least 90% of the SVMs in the EOSVM classifier give the same classification to the DoC group, awake controls, or NotSure class. An agreement from a 90% majority vote is considered to have the credibility to give a medical diagnosis with confidence. It is seen from Fig 9 that approximately 85% accuracy and F1-Score measures are achieved for 6 microstates. This is the best performance among all the settings of microstate maps, indicating the potential of 6 microstates in the diagnosis of DoC.

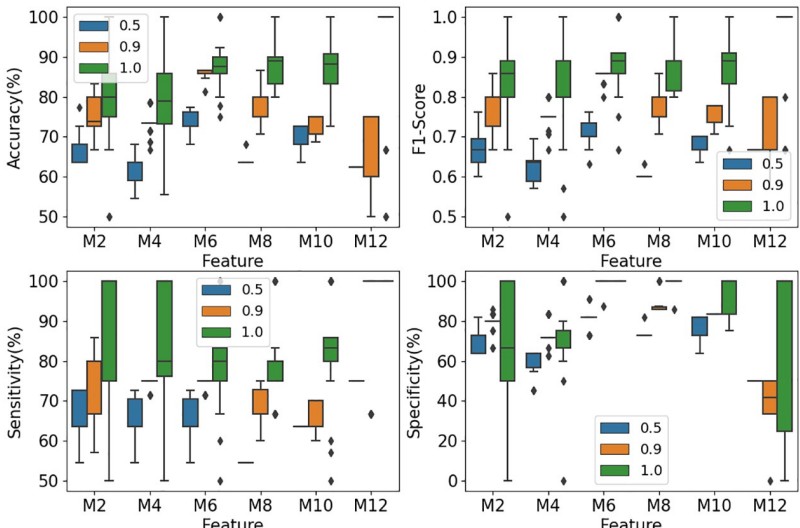

**Fig 9. Classification results from EOSVM.** *M+ number* on the horizontal axis refers to how many microstate maps the features are derived from. For example, M6 refers to the features derived from 6 microstate maps.

The blue box plots in Fig 9 show the classification results when a simple 50% voting rule is applied. It is seen from the figure that all performance metrics are lower than those under 90% majority voting and 100% majority voting. Nevertheless, the performance with the features from 6 microstate maps is still the best. Together with our analysis above on the settings of the number of microstates, we conclude that the setting of 6 microstates is best suitable for the classification of stroke patients.

**EOSVM classification results based on the second group of evaluation metrics.** We evaluated the second group of three metrics defined in Eqs (3), (4) and (5). The evaluation was carried out with a varying number of microstate maps. The number of microstate maps took the values of 2, 4, 6, 8, 10 and 12. The voting rule varied from over 50% majority voting to 100% majority voting of the SVMs in the EOSVM classifier. Some evaluation results are depicted in Fig 10.

With 100% majority voting from all SVMs in the EOSVM classifier, all classifications to DoC or wakefulness were correct for 2, 4, 6, 8 and 10 microstate maps. This was indicated in the TruePrediction results shown in Fig 10(a). The TruePrediction value reached its maximum of 44.84% under 10 microstate maps. The values of metrics in the first group in this situation were also reasonable (with an accuracy of 88.03%, F1-Score of 0.88, sensitivity of 82.61% and specificity of 94.5%). The remaining 55.12% of the patients were classified into the NotSure class, meaning that the classifier was not able to draw a conclusion about whether or not the patients were DoC patients or awake controls. Traditional clinic examinations were needed for the assessment of these patients.

With a 90% majority voting rule, Fig 10(b) shows that the TruePrediction reaches its maximum of 65.91% under 6 microstate maps. In this situation, 34.09% of patients were classified into the NotSure class. Therefore, among the patients who get a clear final prediction, 85.96% of the prediction is true (the accuracy is 85.96%) and the F1-Score is 85.13%.

Fig 10(c) depicts the scenario when an over 50% majority voting rule is applied. All patients could be classified (NotSure = 0). The TruePrediction was significantly high (over 60%). However, the FalsePrediction was also clearly significant (over 30%). Thus, useful information was

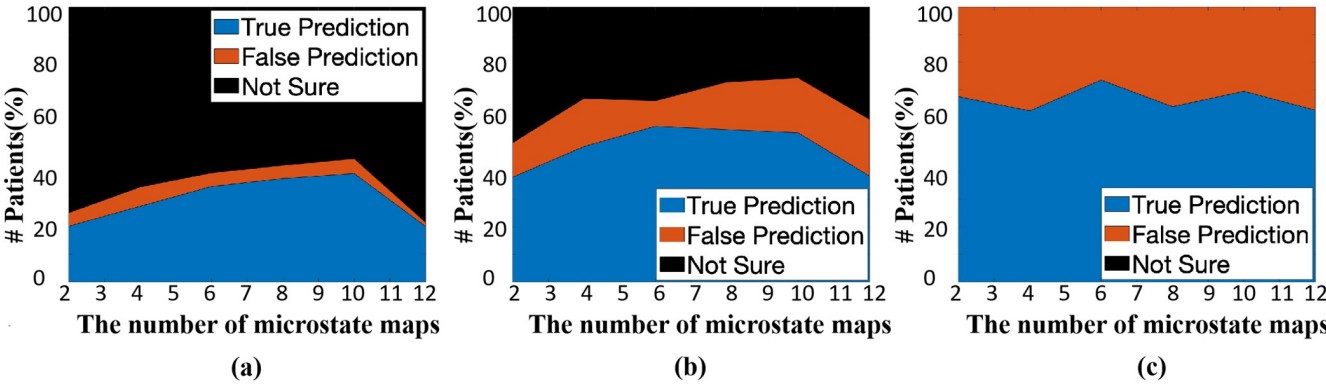

**Fig 10. Evaluation results of TruePrediction, FalsePrediction, and NotSure under different numbers of microstate maps.** (a) represents the 100% majority voting results, (b) represents the 90% majority voting results, (c) represents the 50% majority voting results. The black area represents the proportion of patients that cannot be predicted (NotSure), giving no final prediction from the EOSVM. The red area stands for FalsePrediction from the EOSVM, i.e., either classifies a DoC patient to the wakefulness class or predicts an awake patient to the DoC class. The blue area refers to TruePredictionin percentage, i.e., classifies DoC and awake patients correctly to their respective classes.

not obtained from these results for medical assessments of stroke patients. Thus, a simple over 50% majority voting rule was not applicable in the scenarios that we investigated in this study.

## Further discussions

Our work in this paper makes two main contributions: The first contribution is our establishment of correlations between EEG microstates with the clinical states of stroke patients through statistical analysis. The second contribution is our design of a classifier for the detection of the DoC of stroke patients. We will discuss the statistical analysis and classification, respectively.

### Statistical analysis

EEG microstates reflect the states of consciousness in stroke patients. Through correlation analysis, it has been found that the occurrence, duration, and coverage of microstate F all increase significantly when the consciousness level of a stroke patient becomes worse. In other words, stroke subjects with DoC have higher values of the three parameters of microstate F than those who are awake. Similarly, the duration of microstates A, B and C is higher in the subjects with DoC than in those who are awake. In contrast, the occurrence of microstates B, C, and D are all lower in stroke patients with DoC than in those who are awake. In terms of the coverage of the microstates, microstates A and F show an increasingly dominant position in stroke patients with DoC compared to awake subjects. However, microstate D is dominant in awake subjects.

In comparison with previous studies on consciousness and microstates, there were some similar consequences. When the four canonical EEG microstates in sleep stages and in waking states were compared by Brodbeck and colleagues [30], their results showed that microstate C was dominant in awake states and sleep stages N1 and N3, but microstate B dominated in sleep stage N2. Also, the duration of all four microstates increased in sleep stage N3. Similar to these results, our study showed that the duration of A, B and C increased in subjects who were in a coma compared with those who were awake. This could be partly explained by the incidence of slow waves in the EEG data.

Katayama and colleagues explored changes in EEG microstates in participants undergoing hypnosis [31]. Their analysis demonstrated that the duration and occurrence of microstates B and D decreased during hypnosis relative to rest which supported the notion that microstate D was associated with attention and decreased cognitive control during hypnosis [32]. Similar to these results, our study revealed that the occurrence of microstates B and D decreased in subjects with DoC relative to wakefulness.

EEG microstates also reflect motor disturbance in stroke patients. One-way ANOVA demonstrates that there are significant differences in the occurrence of microstate A and duration of microstate D between the different motor disturbance groups. It is worth noting that the occurrence of microstate A is remarkably higher in the right motor disturbance group than that in the left motor disturbance group, however, the duration of microstate D is significantly higher in the left motor disturbance group than that in the right motor disturbance group. These results show a similar agreement with the report in the pioneering study in [18] on the duration of microstate D in stroke. This previous study showed that in stroke patients, the microstate C and D differed in their duration after both a left and a right lesion concerning controls (C was lower than D in the left, and D was lower than C in the right lesion) [18]. Therefore, the stroke lesions in the brain and motor disturbance after stroke seemed to be both related to the duration of microstate D.

Previous studies also analyzed other related factors of consciousness disorder in stroke patients. For example, Li, Jie, et al. reported that age, stroke severity, and massive cerebral infarct are risk factors for early consciousness disorder [33].

## Classification of DoC patients and awake controls

The dataset in this study is an imbalanced dataset with 57 positive cases (DoC subjects) and 95 negative cases (no DoC subjects). For this kind of heavily imbalanced dataset, common classifiers always predict the minor class as the major class. As a result, the sensitivity of the classification is very low. However, in clinical diagnosis, a test with high sensitivity is necessary for ruling out disease [34]. Sensitivity in this study refers to the ability of the analysis to correctly detect patients with DoC who do have DoC (i.e., true positives).

From Figs 6 and 7, it is seen that the classification results from classifiers SVM and AdaBoost have poor sensitivity performance of lower than 60%. However, the sensitivity from our classifier EOSVM has been improved significantly to above 80%. Therefore, the design of the EOSVM successfully improved the sensitivity of classification in the imbalanced dataset which is useful in clinical applications. Also, the classifier EOSVM in this study improved the detection of the patients with DoC compared with the classifier SVM and AdaBoost.

As this paper focuses on the study of the classification of stroke patients in a hospital, a test with high sensitivity is useful for ruling out disease [34]. Sensitivity in this study refers to the ability of the analysis to correctly detect patients with DoC who do have DoC (i.e., true positives). From Figs 6 and 7, it is seen that the classification results from classifiers SVM and AdaBoost have poor sensitivity performance of lower than 60%. However, the sensitivity from our classifier EOSVM has been improved significantly to above 80%. Therefore, the design of the EOSVM successfully improved the detection of the patients with DoC rather than the awake controls.

The performance of classification between DoC patients and awake controls in the EOSVM demonstrated that microstate parameters were effective features to classify awake controls and DoC patients. We compared the classification results under different numbers of microstate maps and different majority voting rules in the EOSVM. The results showed that when EEG data were clustered into 6 clusters, the EEG parameters contributed to better classification

performance. These results also supported the hypothesis that there might be more than four primary microstates [10, 11].

## Limitation

The EEG data in our study has some limitations. One of the limitations we should mention here is that the EEG data comes from a small number of electrodes (10 electrodes). However, previous studies show that eight or even four electrodes are acceptable in clinical studies [7, 19, 35]. Another limitation is the number of subjects, however previous studies show that for the clinical data, the subject number is acceptable [36, 37]. Our future study will explore EEG data with more electrodes and subjects using deep learning methods and other EEG features.

## Conclusion

Our study explores the relationship between EEG microstates and the clinical states including the consciousness and motor disturbance in stroke patients. Statistical analysis reveals that the occurrence of microstate A and duration of microstate D are correlated with the sides of the motor disturbance in stroke patients. Microstates A, B, C, D, and F are all correlated with the states of consciousness in different parameters. Further classification of stroke patients to DoC or wakefulness from an EOSVM classifier demonstrates that about 65.91% of stroke patients could be predicted with an accuracy of over 85%. Therefore, EOSVM classification based on EEG microstates is helpful to medical practitioners in DoC detection with saved resources that would otherwise be consumed in traditional clinic checks.

## Supporting information

**S1 Data.**
(RAR)

## Author Contributions

**Conceptualization:** Fang Wang.

**Data curation:** Fang Wang, Xueying Zhang, Fengyun Hu.

**Formal analysis:** Fang Wang, Xue Yang.

**Investigation:** Fang Wang.

**Methodology:** Fang Wang.

**Supervision:** Xueying Zhang.

**Writing – original draft:** Fang Wang.

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
