## [Decision Letter · Decision Letter 0]

28 Mar 2023

PONE-D-23-05197Monitoring the after-effects of Ischemic Stroke through EEG MicrostatesPLOS ONE

Dear Dr. Wang,

Thank you for submitting your manuscript to PLOS ONE. After careful consideration, we feel that it has merit but does not fully meet PLOS ONE’s publication criteria as it currently stands. Therefore, we invite you to submit a revised version of the manuscript that addresses the points raised during the review process.

We look forward to receiving your revised manuscript.

Kind regards,

Tommaso Martino, M.D.

Academic Editor

PLOS ONE

Journal Requirements: 

2. Please provide additional details regarding participant consent. In the ethics statement in the Methods and online submission information, please ensure that you have specified what type you obtained (for instance, written or verbal, and if verbal, how it was documented and witnessed). If your study included minors, state whether you obtained consent from parents or guardians. If the need for consent was waived by the ethics committee, please include this information

"This research work was funded by Institutional Fund Projects under grant no. (G: 141-980-1443). Therefore, authors gratefully acknowledge the technical and financial support from the Ministry of Education and King Abdulaziz University, DSR, Jeddah, Saudi Arabia.

Conflicts of Interest: The authors declare that they have no conflicts of interest to report regarding the present study."

"This research work was funded by Institutional Fund Projects under grant no. (G: 141-980-1443). Therefore, authors gratefully acknowledge the technical and financial support from the Ministry of Education and King Abdulaziz University, DSR, Jeddah, Saudi Arabia."

Reviewers' comments:

Reviewer's Responses to Questions

**Comments to the Author**

1. Is the manuscript technically sound, and do the data support the conclusions?

Reviewer #1: Yes

Reviewer #2: Yes

Reviewer #3: Yes

2. Has the statistical analysis been performed appropriately and rigorously? 

Reviewer #1: Yes

Reviewer #2: Yes

Reviewer #3: Yes

3. Have the authors made all data underlying the findings in their manuscript fully available?

Reviewer #1: No

Reviewer #2: Yes

Reviewer #3: No

4. Is the manuscript presented in an intelligible fashion and written in standard English?

Reviewer #1: Yes

Reviewer #2: Yes

Reviewer #3: No

5. Review Comments to the Author

Reviewer #1: The authors investigate the correlation between EEG microstates with the clinical states of stroke patients, such as the level of consciousness, the existence of motor disturbance, and the side of the motor disturbance. This manuscript requires extensive revision before being reconsidered, particularly in terms of the detailed and justification of the proposed method, as well as the completeness of the results and comparisons. Following are my concerns.

1. The foremost concern is number of subjects are very few for experimentation and dataset ethical approval, statistical analysis, age description details are missing.

Did you read studies where authors utilized 60+ subjects to prove subject-independent motor imagery classification? Please see “Towards the Development of Versatile Brain-Computer Interfaces” and “A New Framework for Automatic Detection of Motor and Mental Imagery EEG Signals for Robust BCI Systems. If it looks difficult for you to perform experiments with large number of subjects at least discussed the above-mentioned studies.

2. For EEG Classification, first step is denoising of signals, MSPCA plays vital role, which is a combination of PCA and wavelet. I recommend authors to use MSPCA in discussion,

The details of MSPCA can be found in "Motor imagery BCI classification based on novel two-dimensional modelling in empirical wavelet transform", “Motor Imagery BCI Classification Based on Multivariate Variational Mode Decomposition”, “15.Motor Imagery EEG Signals Decoding by Multivariate Empirical Wavelet Transform-Based Framework for Robust Brain–Computer Interface”, “ Alcoholic EEG signals recognition based on phase space dynamic and geometrical features”

3. For EEG Classification, convolutional neural networks play vital role, thus details should be included in literature, “Alcoholic EEG signals recognition based on phase space dynamic and geometrical features”.

4. Graphical features are one of the newest approaches for identifying underlying patterns of EEG signals and details of these methods can significantly increase the readability. Please include discussion from following articles “Depression Detection Based on Geometrical Features Extracted from SODP Shape of EEG Signals and Binary PSO”, “Recognizing seizure using Poincaré plot of EEG signals and graphical features in DWT domain”.

5. Please provide the details of future direction and possible solutions to continue this topic.

6. Finally, I suggest authors to sit with English native speaker to improve the writing of proposed work.

Reviewer #2: In the current article, the authors of this article designed a framework to automatically detect disorder of consciousness (DoC) in ischemic stroke patients using computer-aided examinations of the electroencephalogram (EEG) microstates. They extracted microstates maps and calculated their statistical parameters to input into classifiers, specifically an ensemble of support vector machines (EOSVM), to identify DoC. The dataset was imbalanced, with the minority of patients being DoC, so EOSVM was designed to address this issue. The experimental results showed that the EOSVM classifier achieved better performance than other classifiers, with an accuracy and F1-Score both higher than 89%, and improved sensitivity from lower than 60% to higher than 80%. The authors concluded that the EOSVM-aided DoC detection based on microstates parameters is useful to medical practitioners and can save resources that would otherwise be consumed in traditional clinical checks.

I want to say thank to authors for their great efforts on the submitted article.

It seems that article is good and is ready to publish, but I just have some recommendations for authors:

1 reference are ok, but is better some of them be updated.

2 the quality of figures are good, but it's better to show fig 1 and 2 in a better way.

3 contribution section is missing in the introduction.

4 please say clearly in introduction the why you are using this method.

5 please see these 3 articles and see how they addressed the results to the problems, I sure am it will help you a lot to improve your manuscript:

A) Defending Smart Electrical Power Grids against Cyberattacks with Deep Q-Learning

B) Model-free tracking control of regular and chaotic trajectories with machine learning

C) Detecting Weak Physical Signal from Noise: A Machine-Learning Approach with Applications to Magnetic-Anomaly-Guided Navigation

D) Adaptive optimal control of unknown discrete-time linear systems with guaranteed prescribed degree of stability using reinforcement learning

Again, thanks a lot for sending your article for our journal to get our comments.

We wait to see the new version of your article soon.

Reviewer #3: The paper presents detection of Ischemic Stroke through EEG Microstates. Major comments are

1. The rationale behind the usage of EEG must be clearly stated. Why not imaging techniques like CT, PET, or MRI which are more accurate? Also, the rationale and motivation of the study is not clear.

2. The literature review is still shallow. The authors are advised to add a table rather than reporting the existing work in text. this will make the manuscript more readable and easy to follow.

3. The authors are advised to perform 10-fold cross validation and LOSO. It would be a good test to check the stability of the performance.

4. Why not use some preprocessing and feature extraction. In the current form, it is not clear how and what features are fed to the classifier.

5. The authors are advised to add confusion matrix with number of signals belonging to each class. It is unclear from the manuscript that how many signals are used for each class. Also, try to present the dataset details using table to have better readability.

6. The authors are advised to compare the results with existing state of the art techniques.

7. The results are poorly presented. The authors are advised to provide interpretation of the results. Also, can authors also provide discussion about the brain regions? Which channels perform better and what brain region has yielded the highest accuracy?

8. Try to compare the results with RF and other ensemble technique to see if it achieves better performance than it.

9. Compare the results of classification with state of the art ML like KNN, decision tree, ANN.

10. Explore the uncertainty and explainability of the results. For e.g. cite

https://www.preprints.org/manuscript/202301.0148/v1

https://www.sciencedirect.com/science/article/pii/S0010482523001415

6. PLOS authors have the option to publish the peer review history of their article (what does this mean?). If published, this will include your full peer review and any attached files.

Reviewer #1: No

Reviewer #2: No

Reviewer #3: No

---

## [Author Response · Author response to Decision Letter 0]

9 May 2023

Thank you for your encouraging comments and suggestions. In our revised manuscript, we have addressed all comments from you and the three anonymous reviewers \\#1, \\#2 and \\#3. According to your request, we have submitted three files named "Response to Reviewers", "Revised Manuscript with Track Changes" and "Manuscript". Thank you.

---

## [Decision Letter · Decision Letter 1]

18 Jun 2023

PONE-D-23-05197R1Monitoring the after-effects of Ischemic Stroke through EEG MicrostatesPLOS ONE

Dear Dr. Wang,

Thank you for submitting your manuscript to PLOS ONE. After careful consideration, we feel that it has merit but does not fully meet PLOS ONE’s publication criteria as it currently stands. Therefore, we invite you to submit a revised version of the manuscript that addresses the points raised during the review process.

We look forward to receiving your revised manuscript.

Kind regards,

Tommaso Martino, M.D.

Academic Editor

PLOS ONE

Reviewers' comments:

Reviewer's Responses to Questions

**Comments to the Author**

1. If the authors have adequately addressed your comments raised in a previous round of review and you feel that this manuscript is now acceptable for publication, you may indicate that here to bypass the “Comments to the Author” section, enter your conflict of interest statement in the “Confidential to Editor” section, and submit your "Accept" recommendation.

Reviewer #2: All comments have been addressed

Reviewer #3: (No Response)

2. Is the manuscript technically sound, and do the data support the conclusions?

Reviewer #2: Yes

Reviewer #3: Partly

3. Has the statistical analysis been performed appropriately and rigorously? 

Reviewer #2: Yes

Reviewer #3: No

4. Have the authors made all data underlying the findings in their manuscript fully available?

Reviewer #2: Yes

Reviewer #3: No

5. Is the manuscript presented in an intelligible fashion and written in standard English?

Reviewer #2: Yes

Reviewer #3: No

6. Review Comments to the Author

Reviewer #2: Thank you for your efforts on addressing all comments

Your article is ready to get published in our journal

Reviewer #3: The authors did not answered all the comments. The authors failed to respond to almost 60% of the comments given in the previous version. The authors are invited to carefully go through the following comments and address it pointwise.

1. The authors have added the some lines about EEG and other modalities.

Resting-state electroencephalography (EEG) monitoring provides an alternative way

to potentially assist medical practitioners in a quick assessment of the after-effects of

stroke. Although the spatial resolution of EEG is lower than images such as CT, PET

and MRI, EEG is noninvasive, easy to monitor long-term and inexpensive, therefore, it

is widely used in neurological examinations. Although the spatial resolution of EEG is

not as good as images such as CT, PET and MRI, EEG recordings always have higher

time domain resolution.

But authors failed to cite any work. f the claims are not the authors original work, it is better to cite them. Please cite and follow the following articles strictly. The authors are recommended to provide wide acceptance of EEG due to their better temporal resolution. Like in drowsiness and emotion detection. In this line following papers are required to be cited:

https://www.sciencedirect.com/science/article/pii/S0003682X21002589

https://www.sciencedirect.com/science/article/pii/S0169260721005241

https://iopscience.iop.org/book/edit/978-0-7503-3279-8/chapter/bk978-0-7503-3279-8ch5

2. Ischemic Stroke: Few lines about this, their statistics around the globe will nicely introduce the problem and their seriousness. The authors are advised to add few statistics about the problem in the first paragraph of introduction.

3. The manuscript will be detailed, if it is compared with existing work on Ischemic Stroke. Try to locate some recent work and compare your results with it.

4. Confusion matrix: The reviewer is not satisfied with the explanation. Please provide the exact number in a Tabular form. The readers do not need to sit with calculator an evaluate the numbers from the Table 1. A separate Table of the actual numbers will make more impact on the manuscript.

5. A one good paragraph on Explainability and Uncertainty will provide readers with future direction. Please add few lines on it. Please cite the papers given in the last version as such works are on the rise because of scope in clinical settings.

https://www.preprints.org/manuscript/202301.0148/v

https://ietresearch.onlinelibrary.wiley.com/doi/full/10.1049/el.2020.2380

https://www.sciencedirect.com/science/article/pii/S0010482523001415

6. Try to strengthen the conclusion with your remarks, findings, and directions.

7. PLOS authors have the option to publish the peer review history of their article (what does this mean?). If published, this will include your full peer review and any attached files.

Reviewer #2: No

Reviewer #3: No

---

## [Author Response · Author response to Decision Letter 1]

11 Jan 2024

First of all, we would like to acknowledge the editor, associate editor and anonymous reviewers for their comments and suggestions. These comments and suggestions are invaluable for us to improve the quality and readability of our manuscript. We have considered these comments and suggestions carefully, and then have revised the paper accordingly. Following the Editor's suggestions, we now resubmit our revised manuscript for your further consideration for publication. The specific response has attached as file.

---

## [Decision Letter · Decision Letter 2]

6 Mar 2024

Monitoring the after-effects of Ischemic Stroke through EEG Microstates

PONE-D-23-05197R2

Dear Dr. Wang,

We’re pleased to inform you that your manuscript has been judged scientifically suitable for publication and will be formally accepted for publication once it meets all outstanding technical requirements.

Kind regards,

Anwar P.P. Abdul Majeed

Academic Editor

PLOS ONE

Additional Editor Comments (optional):

Reviewers' comments:

Reviewer's Responses to Questions

**Comments to the Author**

1. If the authors have adequately addressed your comments raised in a previous round of review and you feel that this manuscript is now acceptable for publication, you may indicate that here to bypass the “Comments to the Author” section, enter your conflict of interest statement in the “Confidential to Editor” section, and submit your "Accept" recommendation.

Reviewer #3: All comments have been addressed

2. Is the manuscript technically sound, and do the data support the conclusions?

Reviewer #3: Yes

3. Has the statistical analysis been performed appropriately and rigorously? 

Reviewer #3: Yes

4. Have the authors made all data underlying the findings in their manuscript fully available?

Reviewer #3: Yes

5. Is the manuscript presented in an intelligible fashion and written in standard English?

Reviewer #3: Yes

6. Review Comments to the Author

Reviewer #3: The authors have addressed all the comments. The paper can be accepted in its current form. Thank you for addressing all the comments.

7. PLOS authors have the option to publish the peer review history of their article (what does this mean?). If published, this will include your full peer review and any attached files.

Reviewer #3: No

---

## [Editor Report · Acceptance letter]

11 Mar 2024

PONE-D-23-05197R2 

PLOS ONE

Dear Dr. Wang, 

I'm pleased to inform you that your manuscript has been deemed suitable for publication in PLOS ONE. Congratulations! Your manuscript is now being handed over to our production team.

Kind regards, 

on behalf of

Dr. Anwar P.P. Abdul Majeed 

Academic Editor

PLOS ONE